# Effects of Probiotics on Gut Microbiomes of Extremely Preterm Infants in the Neonatal Intensive Care Unit: A Prospective Cohort Study

**DOI:** 10.3390/nu14153239

**Published:** 2022-08-08

**Authors:** Ching-Min Chang, Ming-Horng Tsai, Wei-Chao Liao, Peng-Hong Yang, Shiao-Wen Li, Shih-Ming Chu, Hsuan-Rong Huang, Ming-Chou Chiang, Jen-Fu Hsu

**Affiliations:** 1Division of Pediatric Gastrointestinal Disease, Department of Pediatrics, Chiayi Chang Gung Memorial Hospital, Chiayi 613, Taiwan; 2College of Medicine, Chang Gung University, Taoyuan 244, Taiwan; 3Division of Neonatology and Pediatric Hematology/Oncology, Department of Pediatrics, Chang Gung Memorial Hospital, Yunlin 638, Taiwan; 4Molecular Medicine Research Center, Chang Gung University, Taoyuan 244, Taiwan; 5Division of Neonatology, Department of Pediatrics, Chiayi Chang Gung Memorial Hospital, Chiayi 613, Taiwan; 6Division of Neonatology, Department of Pediatrics, Chang Gung Memorial Hospital, Taoyuan 244, Taiwan

**Keywords:** gut microbiome, neonates, probiotics, necrotizing enterocolitis, neonatal immunity

## Abstract

**Background:** Probiotics have been previously reported to reduce the incidence of necrotizing enterocolitis (NEC) in extremely preterm infants, but the mechanisms by which the probiotics work remain unknown. We aimed to investigate the effects of probiotics on the gut microbiota of extremely preterm infants. **Methods:** A prospective cohort study was conducted on 120 extremely preterm neonates (gestational age ≤ 28 weeks) between August 2019 and December 2021. All neonates were divided into the study (receiving probiotics) and the control (no probiotics) groups. Multivariate logistic regression analysis was performed to investigate the significantly different compositions of gut microbiota between these two groups. The effects of probiotics on the occurrence of NEC and late-onset sepsis were also investigated. **Results:** An increased abundance of *Lactobacillus* was noted in neonates who received the probiotics (AOR 4.33; 95% CI, 1.89–9.96, *p* = 0.009) when compared with the control group. Subjects in the probiotic group had significantly fewer days of total parenteral nutrition (median [interquartile range, IQR]) 29.0 (26.8–35.0) versus 35.5 (27.8–45.0), *p* = 0.004) than those in the control group. The probiotic group had a significantly lower rate of late-onset sepsis than the control group (47.1% versus 70.0%, *p* = 0.015), but the rate of NEC, duration of hospitalization and the final in-hospital mortality rates were comparable between these two groups. **Conclusions:** Probiotic supplementation of extremely preterm infants soon after the initiation of feeding increased the abundance of *Lactobacillus*. Probiotics may reduce the risk of late-onset sepsis, but further randomized controlled trials are warranted in the future.

## 1. Introduction

The human gut microbiota is composed of a complicated assembly of aerobic and anaerobic bacteria, viruses, fungi and protozoa, which gradually colonize the neonatal gut immediately following birth [1,2]. The effects of the gut microbiome on human diseases and health have been research interests in recent decades [3,4,5], because the diversity and abundance of gut microbiota maintain human health through the prevention of pathogenic microorganism colonization, modulation of host immune responses and enhancement of nutrients absorbance [6,7]. In neonates, recent evidence has demonstrated significant associations between gut dysbiosis and functional bowel disorders or an increased risk of necrotizing enterocolitis (NEC) [8,9,10].

Various influences from maternal factors, perinatal and dietary exposures, environmental factors and sometimes genetic factors will affect the stepwise process of neonatal gut colonization, which starts very early, soon after birth [11,12]. A strong correlation between gut dysbiosis or the disruption of gut microbiota in early life and metabolic and immune disorders in later life has been documented by various epidemiological studies [13,14,15]. Therefore, researchers have tried to modify gut microbiota activity and composition by dietary interventions such as human milk oligosaccharides, modified formula and, perhaps with more potentiality, by the oral administration of probiotics. Recent systematic reviews and meta-analyses found that the enteral supplementation of probiotics can effectively prevent severe NEC and late-onset sepsis in preterm neonates, but there have been conflicting results [16,17,18]. Additionally, most of the evidence did not investigate the detailed role of probiotics on gut microbiota composition, and the complicated action of probiotics on extremely preterm neonates remains unknown. In this study, we aimed to investigate the effects of probiotics on the development of gut microbiota in very low birth weight infants.

## 2. Methods

### 2.1. Study Design and Participants

A prospective cohort study was conducted and all neonates with a gestational age (GA) of ≤28 weeks and >24 weeks without major congenital anomalies from the NICUs of Chang Gung Memorial Hospital (CGMH) were eligible for enrollment. The NICU of CGMH is a tertiary-level medical center that attends nearly 800 deliveries annually in northern Taiwan. The study was conducted between August 2019 and December 2021. The standard enteral feeding guideline for preterm neonates has been applied in our NICU, and human milk feeding was highly suggested within the first few days for extremely preterm neonates. All eligible neonates who can be successfully weaned from high-setting ventilator support (FiO_2_ > 30) and considered for the initiation of enteral feeding within the first few days of life were approached by our research team. The eligible neonates were divided into the study group (receiving probiotics) and the control group (no probiotics) depending on the decisions of the attending physicians and the family. After the family agreed to participate in this study and signed the informed consent, we would collect a stool sample at the scheduled time. In the study group, neonates had a routine daily dosing of probiotics Infloran capsules (Desma Health care, Chiasso, Switzerland), containing *Lactobacillus acidophilus* and *Bifidobacterium bifidum*, with one capsule (250 mg) once daily soon after the initiation of oral feeding. This study was approved by the Institutional Review Board of CGMH, and written informed consent was obtained from the parents of the participants. All methods were performed in accordance with the relevant guidelines and regulations.

### 2.2. Sample Collection

All fecal samples were collected from infant diapers at 7 to 10 days after the discontinuation of empiric antibiotics and were stored at −20 °C within 10 minutes after collection. Then all samples were transferred to a −80 °C freezer for long-term storage until DNA extraction was performed.

### 2.3. Library Preparation for 16S rRNA Metagenomics Sequencing

DNA was extracted using the DNeasy PowerSoil Kit (cat. 12888-100, Qiagen, Carlsbad, CA, USA). DNA samples were sequenced at the Biotechnology Research Center of the Taiwan University. The 16S rRNA sequencing libraries were prepared according to the manufacturer’s instructions provided by Illumina (Illumina, Los Angeles, CA, USA). Briefly, 12.5 ng of DNA was used for PCR amplification at the V3 and V4 regions of the 16S rRNA gene. The PCR primers contained an overhang adapter sequence followed by the full length primer sequences: Forward: 5′-TCGTCGGCAGCGTCAGATGTGTATAAGAGACAGCCTACGGGNGGCWGCAG, and Reverse: 5′-GTCTCGTGGGCTCGGAGATGTGTATAAGAGACAGGAC TACHVGGGTATCTAATCC. The PCR products were purified with AMPure XP beads (Beckman Coulter, Brea, CA, USA) and subjected to a second PCR reaction with primers from the Nextera XT Index kit (Illumina, USA) to add dual indices and Illumina sequencing adapters. After PCR, the final libraries (~630 bp) were purified with AMPure XP beads and ready for next-generation sequencing.

### 2.4. MiSeq-Based High Throughput Sequencing

The concentrations of 16S rRNA sequencing libraries were determined by real-time quantitative PCR with Illumina adapter-specific primers provided by a KAPA library quantification kit (KAPA Biosystems, Wilmington, MA, USA). Libraries were denatured and sequenced by the Illumina MiSeq platform with reagent v3 for paired-end sequencing (2*250 bp). Instrument control, cluster generation, image capture and base calling were processed by Real Time Analysis software (RTA), MiSeq Control software (MCS) and MiSeq Report software (MSR) on the MiSeq platform. FASTQ files generated by MiSeq Report were used for further analysis. Taxonomic analysis of stool microbes by the V3 and V4 regions of the 16S rRNA sequence was performed using the 16S Metagenomics workflow v1.1.0 with the MiSeq platform, and the classification was based on the Silva database (http://www.arb-silva.de/, accessed on 22 November 2021). Sequence reads were classified at several taxonomic levels: kingdom, phylum, class, order, family, genus and species.

### 2.5. Metagenomics Analysis

DADA2 [19] was used to analyze microbial communities. FastQC was performed to check the read quality. Paired-end sequences were separated through quality filtering, dereplication, denoising, merging and chimera removal based on the DADA2 tutorial. The standard filtering parameters with maxN = 0, truncO = 2, rm.phix = TRUE and maxEE = 2 were used. The truncated forward and reverse sequences were defined at position 240, and the first 13 bases of each sequence were trimmed. A total of 11,799,543 sequences were used to construct amplicon sequence variants (ASVs), and ASVs comprising fewer than ten reads were filtered from the dataset. A Naïve Bayes classifier was trained using the most recent available version of Silva (version 138) sequences for taxonomy assignment for each ASV through the assigned Taxonomy function.

After evaluating the relative abundances of the bacterial taxa in each sample, microbial diversity was then analyzed. Alpha diversity was calculated by the observed ASVs, Shannon index and Fisher’s index with phyloseq [20]. Beta diversity was calculated using principal coordinate analysis (PCoA) with weighted UniFrac distance (*p* = 0.05 by PERMANOVA [21]) using the function “Adonis” of the Vegan package. PICRUSt2 [22] was conducted to predict metagenome functional content from 16S rRNA surveys and full genomes. The predicted genes and their functions were aligned to the METACYC database, and the differences among groups were compared using STAMP (version 2.1.3). A two-sided Welch’s *t* test and Benjamini–Hochberg FDR correction were employed for comparisons of two groups.

### 2.6. Statistical Analyses

Baseline characteristics were compared between the two groups using the chi-square test or Fisher’s exact test for categorical variables, and the Wilcoxon rank sum test for continuous variables. Statistical analyses were conducted using R software (version 3.6.3) (R Development Core Team 2003). The relatively enriched bacteria at the genus level between the probiotic and control groups were investigated by LEfSe analysis [23]. Statistically significant species for each group were evaluated by linear discriminant analysis (LDA) of effect size analysis, which employed the non-parametric factorial Kruskal–Wallis test, the Wilcoxon rank sum test and LDA to identify differential abundant biomarkers between the two groups. An LDA score of higher than 2 or lower than −2 was selected to represent the most significantly enriched genus in these two groups.

## 3. Results

### 3.1. Patients Demographics and Clinical Characteristics

During the study period, a total of 120 preterm neonates were enrolled for analyses (Table 1). The patient demographics including gestational age, birth weight, type of delivery, and most perinatal illnesses were all comparable between these two groups. All of these extremely preterm neonates were prescribed with empiric antibiotics and only 8.3% (n = 10) of them were treated for more than three days due to documented early-onset sepsis. The stool samples were collected at 14.0 (11.0–19.5) days of life, approximately one week after supplementation of probiotics in the study group. The initiation of probiotic supplementation was 8.0 (6.0–12.0) days old in the study group. Most of these neonates (83.3%) were fed with mixed breast/regular formula feeding after their clinical conditions became stable.

All study subjects were followed until discharge or death. There were only 7 neonates who had NEC (≥stage IIa) during hospitalization in the NICU. Although there was no significant difference in the rates of NEC between the probiotic and the placebo groups, the probiotic group had a significantly lower rate of late-onset sepsis than the control group during hospitalization (47.1% versus 70.0%, *p* = 0.015). There were only six in-hospital mortality cases. Additionally, neonates in the probiotic group had significantly fewer days of total parenteral nutrition, (median [IQR]) 29.0 (26.8–35.0) versus 35.5 (27.8–45.0), *p* < 0.005), although the duration of hospitalization was comparable between the two groups. 

### 3.2. Summary of Infant Fecal Microbiota Profiling

Following taxonomic assignment, a total of 11,799,543 qualified sequences (98,330 ± 33,681) and 1310 ASVs were obtained. The median number of reads per infant was 98.3 ± 33.7 and did not differ between the probiotic and placebo groups. The ASVs were classified into known taxa (5 phyla, 9 classes, 28 orders, 51 families, 96 genera, and 84 species) and unclassified groups. The fecal microbial diversities of species in each sample showed significantly higher observed ASVs and Fisher’s indices in the probiotics group (Figure 1). For beta diversity, PCoA plots evaluated by weighted UniFrac distances revealed a significant differential distribution of gut microbiota between the two groups (*p* = 0.005) (Figure 2).

### 3.3. Changes in the Most Abundant Bacterial Taxa in Infants Supplemented with Probiotics

The ten predominant families detected in infant fecal samples were Enterobacteriaceae (mean relative abundance: (59.3%), Bifidobacteriaceae (14.8%), Enterococcaceae (11.3%), Streptococcaceae (2.6%), Veillonellaceae (2.2%), Lactobacillaceae (2.1%), Clostridiaceae (1.3%), Yersiniaceae (1.3%), Staphylococcaceae (0.9%) and Peptostreptococcaceae (0.9%), as shown in Figure 3. The most abundant bacteria at the family level between the two groups were investigated. Among the top ten families, the relative abundance of *Bifidobacteriaceae* and *Lactobacillaceae* were increased and were shown to be significantly different in the administration of probiotics group (16.92% and 3.38%) compared to the abundances in the placebo group (11.75% and 0.3%) (*p* < 0.001). A trend of slightly decreased *Yersiniaceae* and *Staphylococcaceae* abundances was also observed in infants supplemented with probiotics.

Table 2 shows the 16 dominant genera in the fecal samples of the probiotic and control groups. *Enterococcus* and *Klebsiella* were the most prevalent genera in these two groups. Bifidobacterium was the most prevalent genus in specimens from probiotic-supplemented infants (detected in 87% of specimens collected from probiotic infants vs. 46% of specimens collected from control infants), followed by *Lactobacillus*, found in 64% of specimens collected from probiotic infants vs. 28% of specimens collected from control infants. *Enterobacter* was the most prevalent genus in specimens from control infants (84% of the placebo group vs. 64% in the probiotic group). 

### 3.4. Differential Taxa in the Microbiome of the Extremely Preterm Neonates

LEfSe analysis was performed to analyze the relatively enriched bacteria at every taxonomic level among different groups. The cladogram plot shows five taxonomic levels, with the phyla levels and genera levels plotted in the innermost ring and outermost ring, respectively. After probiotic supplementation, the microbiota richness was markedly increased in the probiotic group compared with the placebo group (Figure 4A). The amount of phylum *Actinobacteriota* and class *Actinobacteria* were enriched and had the highest linear discriminant analysis (LDA) score in probiotic subjects (Figure 4B). As shown in Figure 4B and Table 2, the relative abundances of *Bifidobacterium, Raoultella* and *Lactobacillus* were significantly increased in the probiotic group compared with the control group. In contrast, the relative abundances of *Klebsiella, Serratia* and *Staphylococcus* were mostly increased in the control group compared with the probiotic infants. To identify the differential bacterial taxa at the species level, we performed LEfSe analysis based on the known ASVs at the species level. The relative abundances of *Bifidobacterium bifidum*, *Faecalibacterium prausnitzii* and *Enterococcus faecium* were significantly increased and the relative abundance of *Staphylococcus epidermidis* was significantly lower in the probiotic group than in the control group (Figure 4C).

Fifteen genera had a mean abundance of at least 1% in one or both groups (Table 2) and were enrolled in the logistic regression analysis. After adjusting for gestational age and age, only *Lactobacillus* was significantly increased in the probiotic group compared with the control group (adjusted odds ratio 4.33, 95% CI, 1.89–9.96, *p* value = 0.009).

The influences of different feeding type on the gut microbiome of the extremely preterm neonates were also investigated. Although most of our neonates (83.3%) were mixed feeding with breast milk and formula, the standard methods were used for analysis. Based on the composition analyses of the gut microbiome, we did not show significant differences between neonates fed with breast milk, formula and mixed feeding (Appendix A).

### 3.5. Functional Prediction of the Microbiome

To explore the predicted functional capacity of the infants’ fecal microbiota, the functions of bacterial communities were analyzed using PICRUSt2. Bifidobacterium shunt, sucrose degradation IV (sucrose phosphorylase), glycogen biosynthesis I (from ADP-D-glucose), phosphatidylglycerol biosynthesis II (nonplastidic), phosphatidylglycerol biosynthesis I (plastidic) and the superpathway of phospholipid biosynthesis I (bacteria) were associated with the supplementation of probiotics. In contrast, parameters related to preQ0 biosynthesis and menaquinol biosynthesis were increased in the placebo group (Figure 5).

## 4. Discussion

Updated meta-analyses and reviews support the effects of probiotics on preventing NEC and late-onset sepsis in VLBW neonates, reducing the requirement of total parenteral nutrition and reducing mortality and morbidity rates in specific subgroups [18,24,25,26]. Because the incidence of NEC in our cohort was low and only seven neonates developed NEC, we failed to conclude the mechanisms by which probiotics may work to prevent NEC. We found that the early administration of probiotics may potentially reduce the risk of late-onset sepsis and an increased abundance of *Lactobacillus* in probiotic-supplemented infants was observed after logistic regression analysis. This result highlights the importance of *Lactobacillus* when the immature gut of preterm infants develops protective microbiota against the pathological microorganisms that may cause late-onset sepsis.

We found that the probiotic group had a lower incidence of late-onset sepsis in this study. However, this was not a randomized controlled trial and the use of probiotics depended on the decisions of the attending physicians and the family because it has not been the general principle in our institute that probiotics should be routinely prescribed in extremely preterm neonates. Additionally, the timing of probiotic administration, usually after the stable initiation of oral feeding, varied widely in this study. Some of the late-onset sepsis occurred even before the administration of probiotics. Neonates with late-onset sepsis will have a higher risk of recurrent late-onset sepsis [27], which may be a confounder in this study. Current studies have found protective effects of probiotics on reducing late-onset sepsis, but the conclusion remains debatable [7,12,25]. We therefore suggest more prospective, randomized, placebo-controlled trials to examine the beneficial effects of probiotics.

Lactobacillus species are important in the normal flora of healthy term born infants and adult guts [28,29,30]. Previous studies found that *Bifidobacterium* spp. also play an important role in preventing NEC [31,32,33]. In our cohort, *Bifidobacterium* was the second most increased species in the probiotic-supplemented infants. Both Lactobacillus and Bifidobacterium species are known to be significantly fewer in preterm infants, especially in extremely preterm infants [34]. Recent studies supported the concepts that supplementation of probiotics rich in *Lactobacillus* and *Bifidobacterium* early in VLBW infants can help in the development of normal gut flora, accelerating maturation of the immunological responses, influencing the physiological effects of more weight gain, and reducing the bowel permeability and colonization of pathological bacteria, which collaborate together to decrease the risk of NEC and late-onset sepsis [29,35,36,37]. Interestingly, *Klebsiella* spp. and *Staphylococcus* spp. were decreased in the probiotic group, which may imply the effects of probiotics on reducing some pathogenic bacteria [28,32].

Previous studies have implied the importance of achieving early probiotic bacterial gut colonization in low birth weight infants [37]. These studies found that earlier development of the intestinal immune systems and colonization resistance against pathogenic microorganisms can enhance the most beneficial effects of probiotics on NEC protection [29,35,37]. However, some studies concluded that probiotics could not have significant impacts on the incidence of NEC using a multispecies approach [33,38]. The conflicting results may be due to different study designs, cohort selections and different baseline rates of NEC in different institutes [33,39,40,41]. Additionally, there are huge differences between individual *Lactobacillus* and *Bifidobacterium* strains and their abilities to exert immune and infection modulations [39,40,41]. Because of current published studies [29,33,34,35,36,37], our policy was to administer routine daily dosing of *Lactobacillus acidophilus* and *Bifidobacterium bifidum* as early as possible; that is, most of them were administered within the first few days of life. 

Many other factors will affect the immature gut microbiota of preterm neonates, including initial empiric antibiotics soon after birth, patient demographics, nutritional status and nosocomial infections [41,42,43]. In our institute, we followed the AAP (American Academy of Pediatrics) guideline to use empiric antibiotics for extremely preterm neonates [44]. We tried our best to control these variables in this prospective study, and most demographics of the placebo group were comparable to those of the study group. The therapeutic strategies, including the use of total parenteral nutrition, medications for bronchopulmonary dysplasia, and antifungal prophylaxis were all based on our standard protocol. Additionally, we focused on extremely preterm neonates (GA ≤ 28 weeks), who are at high risk of NEC and late-onset sepsis. We suspected that the increased abundance of *Bifidobacterium* would be observed after repeat post-probiotic sampling, which was not included in this study due to some technical problems, including antibiotic use for clinical sepsis, potential confounders of complicated hospital courses, and some therapeutic strategies.

There are some limitations in this study. The sample size of this study is inadequate to conclude the beneficial effects of probiotics on reducing the incidence of NEC. Numerous factors will affect the development of the gut microbiota in preterm infants, and we could not perform a subgroup analysis due to inadequate sample size. Although this was a prospective observational study, the non-randomized study design would inevitably lead to some confounding factors, since agreement of receiving probiotics depended on the decisions of the attending physicians and the family. This was a single center study and approximately one-fourth of all preterm term infants were outborn; thus, some perinatal care and variables were not controlled. Currently, most of the studies can only support an association but cannot definitely conclude the effective impacts of probiotics on reducing the incidence of NEC because unmeasured confounders cannot be completely controlled [37,38]. Therefore, universal probiotic supplementation in preterm infants, especially extremely low birth weight infants, is currently not recommended by the American Academy of Pediatrics [45].

## 5. Conclusions

In conclusion, probiotic supplementation with *Lactobacillus acidophilus* and *Bifidobacterium bifidum* would change the gut microbiota of extremely preterm infants (GA ≤ 28 weeks). Increased abundance of *Lactobacillus* within the first few weeks of life may potentially protect VLBW infants against late-onset sepsis. Further studies are warranted to investigate the impact of probiotics on the microbiota at the strain level, which may explain the mechanisms by which probiotics affect to prevent the occurrence of NEC and/or late-onset sepsis.

## Figures and Tables

**Figure 1 nutrients-14-03239-f001:**
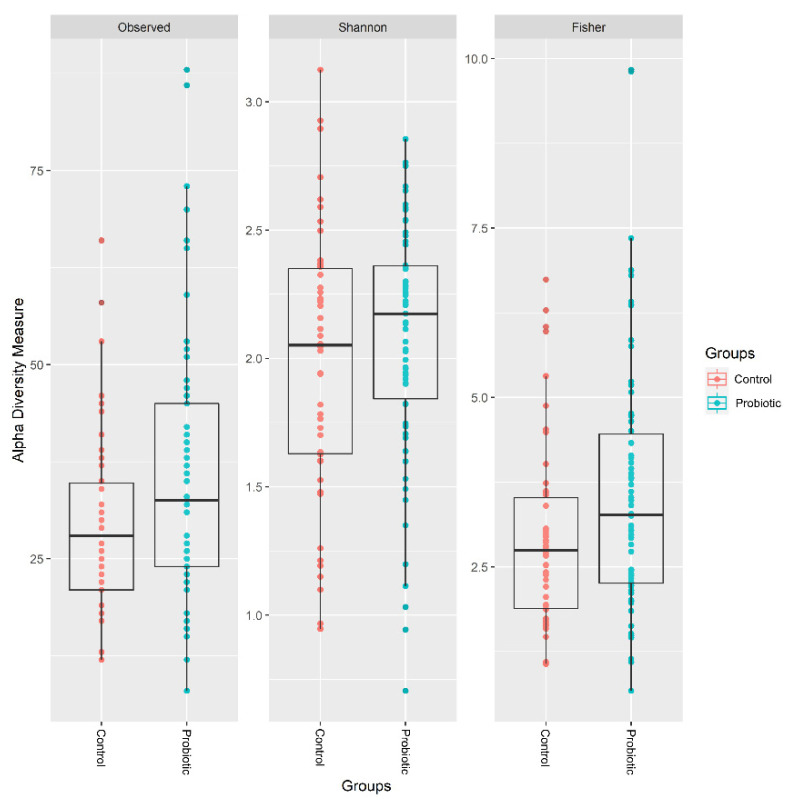
The alpha diversity of the probiotic group was higher than that of the control group using the observed and Fisher diversity indices.

**Figure 2 nutrients-14-03239-f002:**
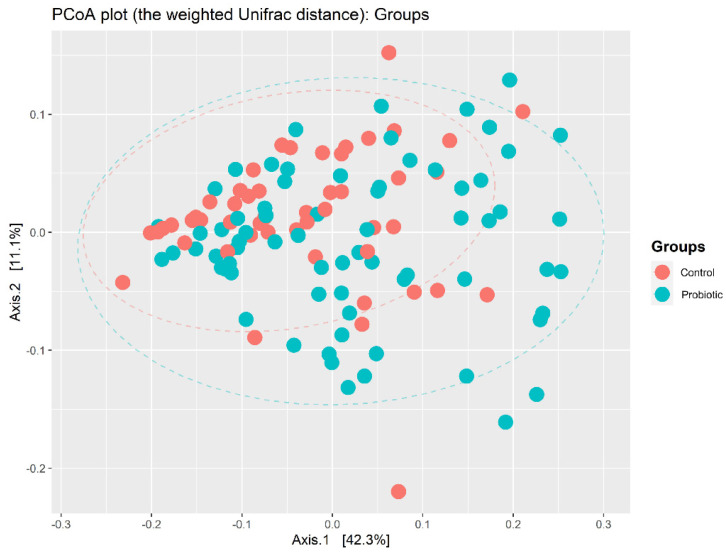
A PCoA plot evaluated by weighted UniFrac distances showed a significant differential distribution of gut microbiota between the probiotic and control groups (*p* = 0.005).

**Figure 3 nutrients-14-03239-f003:**
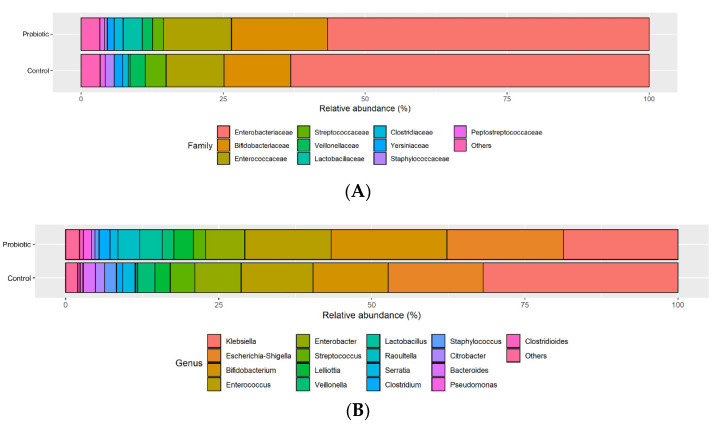
The percentage abundance of specific families (**A**) and genera (**B**) are compared between the probiotic and control infants. The 17 genera included had a mean abundance of at least 1% in one or both allocation groups and were included in the regression analysis. Bacteria less than 1% are grouped as “others”.

**Figure 4 nutrients-14-03239-f004:**
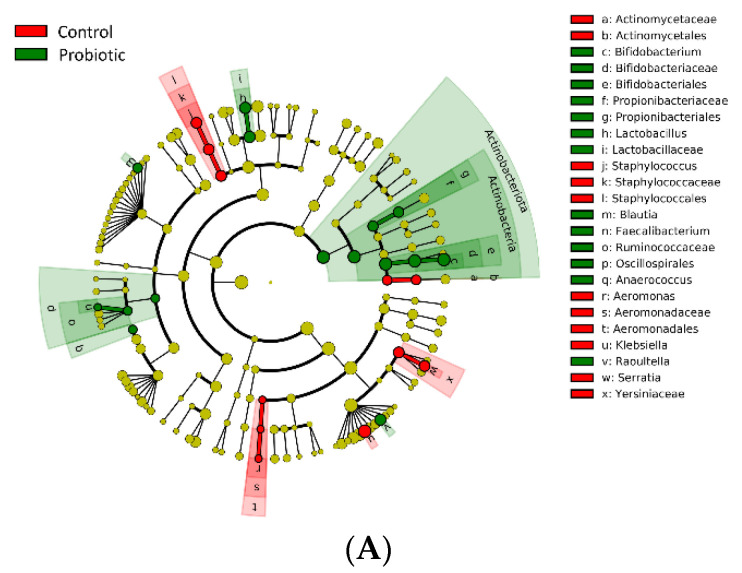
The LEfSe analysis showed that the microbiota richness was markedly increased in the probiotic group compared with the control group (**A**). The cladogram plot shows five taxonomic levels, with the phyla and genera levels plotted in the innermost and outermost rings, respectively. The phylum *Actinobacteriota* and class *Actinobacteria* were enriched and had the highest linear discriminant analysis (LDA) score in the probiotic group (**B**). The relative abundances of *Bifidobacterium bifidum*, *Faecalibacterium prausnitzii*, and *Enterococcus faecium* were significantly increased, but the abundance of *Staphylococcus epidermidis* was significantly lower in the probiotic group than in the control group (**C**).

**Figure 5 nutrients-14-03239-f005:**
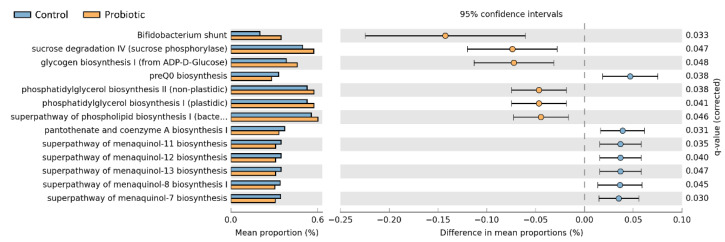
The PICRUSt2 was used to compare functional changes of bacterial communities between neonates with and without supplementation of probiotics. Comparison of the functional profiles between neonates in the probiotic and placebo groups showed that phosphatidylglycerol biosynthesis was slightly enriched in the probiotic group. The preQ0 biosynthesis that is associated with inflammation was also higher in the control group. Bonferroni-adjusted *p*-values < 0.05 indicate significance.

**Table 1 nutrients-14-03239-t001:** The demographics, clinical features and outcomes of neonates in the study group versus the placebo group.

	The Probiotic Group (Total n = 70)	The Control Group (Total n = 50)	*p* Values
Patients demographics			
Birth body weight (g), median (IQR)	780.0 (689.3–915.0)	815.0 (757.5–920.0)	0.511
Gestational age (weeks), median (IQR)	26.0 (25.0–27.0)	26.0 (25.0–27.0)	0.621
Gender (male/female)	36 (51.2)/34 (48.6)	32 (64.0)/18 (36.0)	0.194
NSD/Cesarean section	18 (25.7)/52 (74.3)	19 (38.0)/31 (62.0)	0.166
Inborn/outborn	12 (17.1)	10 (20.0)	0.812
5 minutes Apgar score ≤ 7, n (%)	25 (35.7)	11 (22.0)	0.157
Perinatal asphyxia, n (%)	7 (10.0)	5 (10.0)	1.000
Premature rupture of membrane, n (%)	27 (38.6)	25 (50.0)	0.263
Intraventricular hemorrhage (≥ Stage II), n (%)	10 (14.3)	8 (16.0)	0.801
Initial use of antibiotics			
Ampicillin plus gentamicin, n (%)	38 (54.3)	25 (50.0)	0.712
Ampicillin plus cefotaxime, n (%)	32 (45.7)	25 (50.0)	0.712
Early-onset sepsis, n (%)	5 (7.1)	5 (10.0)	0.740
Duration of initial empiric antibiotics (days), median (IQR)	3.0 (1.0–7.0)	2.0 (1.0–5.0)	
Feeding, n (%)			0.286
Breast feeding	5 (7.1)	4 (8.0)	
Regular formula feeding	4 (5.7)	7 (14.0)	
Mixed (breast feeding plus regular formula feeding)	61 (87.1)	39 (78.0)	
Day of feeding initiation (day), median (IQR)	5.0 (3.0–9.0)	4.0 (3.8–7.0)	0.555
Day of stool sample collection (day), median (IQR)	14.0 (11.0–19.0)	14.0 (11.0–20.0)	0.487
Final outcomes, n (%)			
Necrotizing enterocolitis (≥ stage II)	5 (7.1)	2 (4.0)	0.469
Late-onset sepsis	33 (47.1)	35 (70.0)	0.015
Duration of total parenteral nutrition/Intrafat (days), median (IQR)	29.0 (26.8–35.0)	35.5 (27.8–45.0)	0.004
Duration of hospitalization (days), median (IQR)	96.5 (88.0–112.0)	98.0 (89.0–116.8)	0.269
In-hospital mortality	3 (4.3)	3 (6.0)	0.535

IQR: interquartile range.

**Table 2 nutrients-14-03239-t002:** Logistic mixed-model regression analysis for examining the effect of probiotic supplementation on bacterial genera abundance.

Genus *	Probiotic (n = 70)		Control (n = 50)		AOR ** (95% CI)	*p* Value	Adjusted *p* Value
	Prevalence n (%)	Relative Abundance Mean% (SD)	Prevalence n (%)	Relative Abundance Mean% (SD)			
*Bifidobacterium*	61 (87)	18.9 (20.5)	23 (46)	12.2 (19.0)	1.71 (0.83–4.03)	0.22	0.32
*Enterobacter*	45 (64)	6.4 (16.2)	42 (84)	7.6 (18.7)	0.42 (0.19–0.93)	0.031	0.124
*Escherichia/Shigella*	48 (69)	19.1 (23.8)	29 (58)	15.5 (24.2)	1.99 (0.90–4.43)	0.091	0.183
*Klebsiella*	62 (89)	18.7 (25.6)	46 (92)	31.8 (30.2)	0.32 (0.14–0.76)	0.01	0.055
*Staphylococcus*	51 (73)	0.6 (1.5)	41 (82)	1.9 (4.4)	0.35 (0.15–0.78)	0.01	0.055
*Enterococcus*	69 (99)	14.1 (14.7)	48 (96)	11.8 (14.7)	1.51 (0.73–3.14)	0.27	0.329
*Streptococcus*	50 (71)	2 (5.6)	35 (70)	4 (9.3)	0.62 (0.28–1.38)	0.24	0.327
*Veillonella*	26 (33)	1.9 (5.2)	14 (28)	2.9 (8.9)	1.14 (0.47–2.75)	0.77	0.767
*Clostridium*	19 (27)	1.8 (6.6)	15 (30)	1 (2.8)	0.38 (0.13–1.10)	0.073	0.167
*Lactobacillus*	45 (64)	3.5 (7.3)	14 (28)	0.3 (1.3)	4.33 (1.89–9.96)	0.001	0.009
*Citrobacter*	23 (33)	0.6 (1.6)	15 (30)	1.5 (4.6)	0.71 (0.28–1.83)	0.48	0.514
*Bacteroides*	26 (37)	0 (0)	11 (22)	2 (12.6)	2.02 (0.85–4.75)	0.11	0.195
*Serratia*	26 (37)	1.3 (6.5)	28 (56)	2 (7.1)	0.46 (0.22–0.97)	0.042	0.132
*Raoultella*	19 (27)	3.6 (12.8)	6 (12)	0.1 (0.4)	2.73 (1.00–7.45)	0.049	0.132
*Lelliottia*	41 (59)	3.2 (9.9)	21 (42)	2.5 (7.1)	1.74 (0.83–3.62)	0.14	0.224

* Proportional abundances of each genus were converted to a binary variable based on the median value. Only genera that had a mean abundance of at least 1% abundant in one or both allocation groups were included in the regression analysis. ** Odds ratio for mixed effects regression model association between the allocation group and bacterial abundance adjusted for gestation and age at sampling, clustering by participant number to account for multiple specimens from infants. AOR: adjusted odds ratio; 95% CI: 95% confidence interval.

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
