# Peer review of "Effects of Probiotics on Gut Microbiomes of Extremely Preterm Infants in the Neonatal Intensive Care Unit: A Prospective Cohort Study"

_nutrients, 2022, doi:10.3390/nu14153239_

Round 1

Reviewer 1 Report

 The authors have taken into account the suggested revisions. Only one sentence in the conclusion would need to be changed: "Increased abundance of Lactobacillus within the first few weeks of life may potentially protect VLBW infants against NEC". This study does not conclude that, it does conclude that :Increased abundance of Lactobacillus within the first few weeks of life may potentially protect VLBW infants against late-onset sepsis.

Author Response

Dear reviewer:

     Thank you for your reviews and comments. Please see the attachment.

Best regard,

Tsai Ming Horng

Reviewer 2 Report

The authors aimed to evaluate the role of probiotics correlated with gut microbiomes and clinical significance. The authors did valuable research to analyze the data from the prematurity. There are some points necessary to be clarified:

Author Response

Dear reviewer:

      I appreciate your review and comments. Please see the attachment.

Best regard,

Tsai Ming Horng

Round 2

Reviewer 2 Report

I reviewed your revised version. I'd appreciate you accept and consider my advice. 

This manuscript is a resubmission of an earlier submission. The following is a list of the peer review reports and author responses from that submission.

Round 1

Reviewer 1 Report

The authors aimed to evaluate the role of probiotics correlated with gut microbiomes and clinical significance. The authors did valuable research to analyze the data from the prematurity. There are some points necessary to be clarified:

  1. The kind of feeding is one of the most important factors influencing gut microbiomes. Although the differences of feeding type were not significant between two group, it is necessary to analyze the difference in gut microbiome according to kind of feeding.
  2. There have been very well reported about studies on gut microbiota in preterm infants related with severe diseases. But your discussion supporting your study seemed too weak.

Author Response

Dear reviewer:

      Please see the attachment. I appreciate your comments, thank you.

Best regard,

Tsai Ming Horng

Reviewer 2 Report

The authors describe a fairly large trial in preterm infants targeting modifications in gut microbiota using a commercially available probiotic. It is unclear how an infant was assigned to either the probiotic or control groups, apparently this wasn't randomized but by decision "of the attending clinician and the family". Did 50 families agree to participate but not take the probiotic? Did these preterm infants really swallow a large capsule containing 250mg probiotics or was the capsule opened and contents dissolved? What was the placebo? If none was used be clear about that! It is also unclear when probiotic was administered as authors state that even some of the late onset sepsis occurred BEFORE probiotics were given. This needs to be clarified! However, without providing exact timing of stool sample collection it is not clear if probiotics were even given before the stool sample was collected. The methods for the microbiota analysis are appropriate. While it is to be expected that probiotic supplementation with a product will increase the amounts of bacteria contained in the probiotic (LAB, Bifido) the observation that in the control group Klebsiella and Staph were increased is interesting.  As incidence of late term sepsis is different between the two groups, there should be some reference to the relevance of this condition in the introduction and some discussion. The potential increase in NEC in the probiotic group, not reaching significance here potentially due to limited statistical power, should be discussed especially given the non randomized design. The current conclusion that probiotic supplementation IMPROVED gut microbiota needs to be rephrased, as all the authors show is that providing a probiotic containing LAB increased LAB. If that is good or bad remains to be established.

Author Response

(The authors gave the same response as above.)

Reviewer 3 Report

Abstract:

"The effects of probiotics on the occurrence of NEC and late-onset sepsis were also investigated": incidence of NEC is very low now. With only 120 neonates included, it is obvious that authors could not find any significative difference between the 2 groups, concerning NEC occurence.

Introduction:

In the first sentence, we understand that intestinal colonization begins after birth but in the sentence line 48 to 50, we understand that it begins very early during fetal life. Perhaps authors should clarify this point. The paradigm of a sterile uterine environment was challenged by the paper of Aagaard et al published in 2014. But since then, these results have been criticized (de Goffau, Nature 2019).

Line 55-57: It is important to remember that in these studies, most preterm infants are fed formula. Mother's milk is the best nutritional strategy for newborns. Studies comparing preterm infants fed with mother's milk versus preterm infants fed with formula + probiotic or mother's milk supplemented with probiotic would be more interesting. In fact, as authors mentioned later, most neonatal units recommend human milk feeding.

Methods= when supplementation is starded, first day of enteral feeding?

Results

Line 145: Why all neonates were prescribed with empiric antibiotic? What are recommendation in this center concerning early onset sepsis? Reducing exposition to antibiotic in the first day of life is really a challenge and can modify neonatal intestinal colonization.

Table 1 and 2 are not included in this manuscript.  I can’t do it without important informations included in Table 1 and 2.

Author Response

(The authors gave the same response as above.)

Round 2

Reviewer 2 Report

The author have done a good job addressing concerns. However, there continues to be some confusion about the study design. While the authors refer to this as a prospective cohort study, such a study would be analyzed between exposed and unexposed individuals but not divided into intervention and controls. The design used here appears to be a non-blinded, non-randomized intervention study. Clearly there was some interference by the clinician in deciding about probiotic use as well as instructions on what, when and how to use it. Thus, it is clearly not an observational but an intervention study. As such the authors need to provide a strong argument of why they did neither blind nor randomize. There might be some value to the reported results despite the poor study design, but the authors need to be clear about the actual design and their reasons for not using a more appropriate (double blinded, randomized) intervention study design.

Author Response

Dear reviewer:

       Please see the attached file. I appreciate your review and comments, thank you.

Best regard,

Tsai Ming Horng

Reviewer 3 Report

The authors have made a real effort to make the method and results of this study more understandable.

The method of this study, although not randomized, more or less allows to conclude that there are differences in microbiota between the 2 groups.

However, the method of this study is very questionable concerning the secondary objectives (occurrence of NEC and late sepsis). Indeed, the fact of having a non-randomized distribution of the children between the probiotic group and the control group, left to the appreciation of the clinician, leads to the possibility of numerous confounding factors that should have been detailed (percentage of mother's milk in the enteral feeding of the newborn at the time of stool sampling, time of probiotic supplementation, median time of occurrence of the episode of late sepsis, number of children with enteropathy in each group (NEC stage I).

Methods:

“with relatively stable conditions judged 71 by the attending physicians”: this sentence suggests that the inclusion criteria are unclear, which prevents the reproducibility of this study

Results:

Summary of clinical characteristics: in this chapter there is no description of clinical characteristic, it is in the previous chapter.

What is the median age for probiotic supplementation?

At stool collection, what is the median proportion of mother's milk in the newborn's enteral diet in each group?

Differential taxa in the microbiome of the effects of probiotics: sentence to rephrase

PAGE 8 LINE 228 “and the of Staphylococcus epidermidis was significantly lower in the probiotic group than in 228 the control group (Figure 4C).” a word is missing at the beginning

Table 2 does not add much to the results reported in Figure 4B. It does not seem necessary to keep it

Page 9 lines 244-246: “The relatively enriched bacteria at the genus level between the probiotic and control groups were investigated by the LEfSe analysis. An LDA score of higher than 2 or lower than -2 was selected to represent the most significantly enriched genus in these two groups.” This sentence shoud be in the Methods chapter

Page 9 lines 247-250: “As shown in Figure 4B and Table 2, the relative abundances of Bifidobacterium,  Raoultella, and Lactobacillus were significantly increased in the probiotic group compared  with the control group. In contrast, the relative abundances of Klebsiella, Serratia and  Staphylococcus were mostly increased in the control group compared with the probiotic  infants.” This sentence should be inserted line 244, before the description of figure 4C.

Page 9 line 258: supplemental files: it is necessary to put legends for the additional figures.

Page 10 line 284 285: “This result highlights the importance of Lactobacillus when the immature gut of preterm infants develops protective microbiota against the occurrence of NEC or late-onset sepsis=> It is not true for the occurrence of NEC

Author Response

(The authors gave the same response as above.)
